# Self-Expandable Metal Stents for Obstructing Colon Cancer and Extracolonic Cancer: A Review of Latest Evidence

**DOI:** 10.3390/cancers17010087

**Published:** 2024-12-30

**Authors:** Pedro Marílio Cardoso, Eduardo Rodrigues-Pinto

**Affiliations:** 1Gastroenterology Department, Centro Hospitalar Universitário de São João, 4200-319 Porto, Portugal; 2Faculty of Medicine, University of Porto, 4200-319 Porto, Portugal

**Keywords:** colon cancer, self-expandable metal stents, obstruction, bridge-to-surgery, palliative interventional endoscopy

## Abstract

Colorectal cancer is a leading cause of cancer-related mortality, with a significant percentage of patients presenting with malignant colorectal obstruction (MCO). Self-expandable metal stents (SEMSs) have become an essential intervention in this setting. This review examines the latest evidence regarding SEMS use, discussing indications, contraindications and its efficacy in both curative and palliative care settings. We explore the current evidence of SEMS placement in left- and right-sided MCO, with a particular focus on long-term oncological outcomes, which have been a point of debate in clinical guidelines, as well as the safety of chemotherapy during stent-in-place. Success in SEMSs use is highly dependent on technical skill and patient selection. This review highlights the necessity of a multidisciplinary approach for improving outcomes in SEMS placement for MCO.

## 1. Introduction

Colorectal cancer (CRC) is the third most frequently diagnosed malignancy in the world and the second cause of cancer-related mortality [1]. Approximately 7% to 29% of patients with CRC experience malignant colorectal obstruction (MCO) [2,3], usually at the level of the sigmoid colon, with 75% of tumors located distal to the splenic flexure [4,5]. However, MCO may also occur from advanced intra-abdominal extracolonic malignancies or peritoneal carcinomatosis, either due to extrinsic compression or infiltration of the lumen.

When managing acute MCO, a multidisciplinary evaluation is essential, with close collaboration among interventional endoscopists, surgeons, radiologists, and oncologists to ensure the best optimal patient outcomes [6]. Historically, the standard treatment for MCO was emergency surgery (ES), typically associated with high morbidity and mortality rates [7]. The treatment goals for tumor obstruction should be to prevent immediate complications, optimize tumor control, and facilitate prompt clinical recovery [8].

Since the first report of a self-expandable metal stent (SEMS) placement for the palliation of MCO [9], numerous studies have evaluated the efficacy of SEMSs in managing MCO. Multiple randomized controlled trials (RCTs) as well as several guidelines have been published on colonic stenting for neoplastic obstruction [4,7,8,10,11]. While SEMSs are associated with good short-term outcomes in experienced centers, long-term results, specifically recurrence and survival, are still a subject of debate [12]. Therefore, their use in specific clinical contexts deserves further discussion [4,7].

This review aims to provide a comprehensive and updated overview of the use of SEMSs for MCO. This covers indications and contraindications for stent placement in both the right and left colon, explores its role as a bridge-to-surgery (BTS) strategy, addresses cases of extra-colonic obstruction, and reviews the technical considerations and potential adverse events (AEs) associated with SEMS placement.

Figure 1 presents a visual algorithm designed to guide readers through the indications, contraindications, and procedural steps for SEMS placement in MCO management. It illustrates key decision-making points and emphasizes essential considerations at each stage.

## 2. Clinical Aspects, Indications and Contraindications

Patients typically present at the emergency department with an absence of bowel movements, abdominal pain and distension, nausea and vomiting [5]. If MCO is suspected, a contrast-enhanced CT scan should be performed. This often identifies the cause of the obstruction, pinpoints the location of the stenosis, and provides insights into the local staging of the disease [5,11,13]. CT imaging is also important to identify the presence of abdominal complications that may require surgical intervention and contraindicate colonic stenting.

Colonic stenting should only be performed for patients with obstructive symptoms and radiological evidence of obstruction [6,7]. The primary goal is to relieve obstruction, allowing the resumption of normal bowel function, reducing colonic distension and preventing complications such as necrosis and perforation [4,12,14,15].

The only absolute contraindication to SEMS placement is the presence of perforation, even though indirect signs of colonic ischemia, like parietal pneumatosis, may suggest against stent placement. Other clinical scenarios like peritoneal metastases or tumors located within 5 cm of the anal verge should be carefully considered, as the procedure tends to be less successful in these situations [7]. Moreover, the use of SEMSs in low rectal cancer has been linked to chronic pain, tenesmus and stent migration [4]. When considering stenting longer strictures (greater than 5 cm), caution is also advised as it may reflect a benign lesion, such as diverticulitis, which is associated with a higher risk of perforation. Some studies suggest that longer strictures associated with locally advanced tumors might benefit from prior induction therapy [7], reinforcing the importance of urgent multidisciplinary consultation with the different specialties involved in the management of the patient before considering stenting. While stenting proximal colonic strictures may be associated with lower clinical success rates [16], recent data show it is safe and feasible if performed by a competent endoscopists [17].

## 3. Left-Sided Colon Cancer—Bridge-to-Surgery

When considering stent placement as a BTS for left-sided MCO, there are several aspects to take into consideration. By allowing the relief of obstruction, stenting allows for elective surgery to be performed, permitting pre-operative bowel cleansing, full pre-operative staging, and potentially performing a one-stage surgery.

While short-term outcomes favor stent placement (lower morbidity rates, similar postoperative mortality rates, reduced risk of permanent stoma and higher rates of primary anastomosis) [15,18], the major concern is the oncological outcome, particularly the risk of tumor recurrence. While this risk is specially increased in the case of an occurrence of perforation during stent placement, this risk may also be increased even in patients with uneventful procedures. Theoretically, the shear forces generated by SEMSs could increase the interstitial pressure as well as circulating tumor cells, leading to the development of more aggressive pathological features following SEMS placement. This might increase recurrence rates and negatively impact survival [19].

These concerns initially led to limited recommendations for SEMS placement in this context, as highlighted in the 2014 European Society of Gastrointestinal Endoscopy (ESGE) guidelines, which did not endorse it as the standard treatment. Instead, stent placement was recommended only for high-risk patients, such as those with an ASA score of III or higher and/or age over 70 years, as an alternative to ES [16]. Based on several studies supporting the advantages of SEMS over ES in the short term, with long-term oncological outcomes remaining a concern, this issue was later revised in the 2020 ESGE guidelines, leading to the recommendation that treatment should be decided through a shared decision-making process. Essentially, the recommendation was that for younger patients, surgery might be the preferred option, while for older patients, with more comorbidities, BTS could be an acceptable approach [7]. Both the 2021 American Gastroenterological Association (AGA) [11] and the 2022 American Society of Colon and Rectal Surgeons guidelines [8] made similar recommendations, stating that SEMS placement is a reasonable choice to allow for one-stage, elective resection [11], although highlighting the potential role of a diverting colostomy with interval colectomy [8].

These guideline recommendations are mainly based on research that used suboptimal methodologies and was often hampered by high colonic perforation rates. More recently, several systematic reviews and metanalyses with more robust methodologies have been published (Table 1 and Table 2). In 2020, a meta-analysis with almost 4000 patients compared SEMS and ES, and concluded that colonic stenting resulted in improved surgical and short-term outcomes, with no difference in long-term oncological outcomes, presented as 3- and 5-year disease-free and overall survival rates [18]. Similar results were reported by the 2020 ESCO trial, a big multicenter RCT evaluating long-term oncologic outcomes between BST and ES, with no significant differences in overall survival, time to progression, or disease-free survival between the two groups after a minimum follow-up of 3 years [20].

In 2021, a Japanese multicenter study also presented favorable long-term outcomes, with 5-year overall survival and relapse-free survival rates of 67.4% and 57.9%, respectively. The overall recurrence rate was 31.0%, with higher risks associated with perforation, although perforation was rare [22]. In 2022, the RCT CReST study also showed no significant differences in recurrence over 3 years after treatment [21], a time frame within which the seeding of tumor cells would be expected to become clinically apparent. These long-term outcomes were further supported by a 2023 Bayesian meta-analysis [23].

It is important to note that all these studies reported high clinical success rates. In general, technical success was reported in >95% of cases with a low AEs rate, including perforation (reported between 1.9 and 3.9%). However, experience and procedural volume are crucial factors that significantly influence the long-term outcomes of SEMS placement, with higher volumes and greater expertise associated with improved technical success and better oncological results. Therefore, a decompressing stoma such as BTS may be considered when SEMS placement expertise is not available [7].

### 3.1. Safety of Chemotherapy During Stent Therapy in the Curative Setting

Regarding the role of chemotherapy in the curative setting as a BTS, it is important to recognize that locally advanced CRC has a high risk of distant disease estimated at 20–30%; therefore, adjuvant and neoadjuvant chemotherapy is often indicated [13,24].

Based on systematic reviews and meta-analysis, neoadjuvant chemotherapy has been shown to be safe and effective in downstaging tumors, thereby increasing the likelihood of achieving complete resections, without increasing the rate of complications, making it a viable preoperative strategy [25,26]. While the FOxTROT trial [27] clearly showed a significant decrease in R1 resection rate in patients who received neoadjuvant chemotherapy for locally advanced colon cancer, only a few patients in this trial underwent SEMS placement as BTS; therefore, the safety of administering chemotherapy after stent placement is still questionable.

In 2023, one comparative study involving 100 patients investigated the safety of neoadjuvant chemotherapy after SEMS placement. There were no statistically significant differences in stent-related complications, although the complication rate was slightly higher in the stent group. In patients who received stent placement followed by neoadjuvant chemotherapy, there was a higher rate of lymph node resection (25.6 vs. 21.8; *p* = 0.04), a more frequent use of laparoscopic surgery (77.1% vs. 40.4%; *p* < 0.001) and a reduced need for stoma creation (10.4% vs. 28.8%; *p* = 0.02) [28] (Table 1).

In 2024, a Japanese multicenter retrospective study with 129 patients with stage II/III CRC concluded that adjuvant chemotherapy significantly improves relapse-free survival after BTS stent placement. The results show that the 3-year relapse-free survival rate was significantly higher in the adjuvant chemotherapy group compared to the no-adjuvant chemotherapy group (78.5% vs. 56.4%; *p* = 0.003) [29] (Table 2).

Even though further research is needed, the available evidence suggests a potential benefit of neoadjuvant and adjuvant chemotherapy after BTS stent placement.

### 3.2. Time Interval Between BTS Stenting and Surgery

Regarding the timing of surgery after stent placement, there are several aspects that should be taken into consideration. On one hand, allowing more time for the patient to recover after stent placement can reduce the likelihood of surgical complications by optimizing the patient’s clinical condition, which can lower the risks associated with subsequent surgical resection. On the other hand, extending this time increases the risk of stent-related complications, particularly the risk of recurrence, which remains the primary concern.

Stent-related AEs, such as perforation, tend to occur within the first seven days, meaning that reducing the interval to surgery would likely not prevent these early complications. There is evidence that delaying surgery by 10 to 15 days significantly reduces the risk of postoperative complications, such as anastomotic leakage, and improves surgical outcomes. Based on the available data, the 2020 ESGE guidelines recommended a time interval of approximately two weeks before surgery [7].

Recent studies have supported this two-week bridging interval as appropriate, as it has been associated with lower stoma formation rates and higher rates of laparoscopic surgery, without any significant differences in short-term and long-term outcomes, including surgical complications, mortality, and survival [32,33,34]. A recent review suggests that extending the interval between SEMS insertion and elective surgery, along with the administration of neoadjuvant chemotherapy, may improve both surgical outcomes and long-term survival [35]; however, further research is needed to better understand the impacts of neoadjuvant chemotherapy after colonic stenting and surgery timing.

## 4. Palliative Setting

In the palliative management of malignant colon obstruction, the use of stents is widely supported across most clinical guidelines, with little debate surrounding its efficacy [7,8,11]. Colonic stenting is endorsed as the preferred option in the palliative setting, as it provides significant relief from obstructive symptoms and reduces the need for invasive procedures. While there is a potential for more long-term complications in patients who survive longer, it is important to note that these patients typically have a short life expectancy [36,37].

Stenting is associated with shorter hospitalization, a lower intensive care unit admission rate, shorter time to initiation of chemotherapy and lower surgical stoma formation (Table 1 and Table 2). The reported clinical success rates are high and the overall complication rates are manageable, making SEMSs a valuable tool in improving patient outcomes and quality of life in palliative care [30,38,39].


*Safety of Chemotherapy in the Palliative Setting*


The impact of chemotherapy in the colonic stenting presents a complex challenge. By prolonging survival, there is an increased risk of stent-related AEs, namely, migration (due to tumor regression) or obstruction (due to tumor in/overgrowth). However, the main concern of integrating chemotherapy, especially with antiangiogenic agents like bevacizumab, into the treatment regimen post-stent placement is the potential for an increased risk of perforation. This risk arises from the synergistic effects of chemotherapy and the mechanical pressure exerted by the stent on the tumor. Bevacizumab is a vascular endothelial growth factor inhibitor that can lead to tumor regression and necrosis, which may weaken the bowel wall. When coupled with the radial pressure from the stent, this weakening can increase the risk of perforation. Even though no data are available regarding newer antiangiogenics agents like regorafenib or aflibercept, they are speculated to have a similar risk to bevacizumab.

While the literature is consistent on the benefits of initiating chemotherapy without antiangiogenic agents in patients who have undergone palliative stenting [7,40], data still remain limited and contradictory on the risk of stent-related perforation after the initiation of first-line bevacizumab-based systemic therapy [41,42,43]. Bevacizumab is part of the standard first-line systemic therapy in metastasized CRC because of its positive effect on response rate, progression-free survival, overall survival, or a combination of these results depending on the combined chemotherapy regimen. In clinical practice, the potential risks of future bevacizumab therapy should not exclude endoscopic stent placement in the emergency treatment of life-threatening acute MCO, particularly not when patients have an increased risk of mortality after ES owing to high age or severe comorbidities, and when local expertise with colonic stent placement is available. ESGE guidelines recommend that chemotherapy with antiangiogenic drugs can be considered for patients following SEMS placement, but advise against colonic stenting in patients while receiving it [7]. In the later cases, extrapolating results from surgical societies, a minimum of 6 to 8 weeks between the last dose of bevacizumab and stent placement should be allowed.

## 5. Proximal/Right-Sided Colon Cancer

Regarding stent placement in the proximal colon (proximal to splenic flexure), the 2014 ESGE guidelines favored surgical intervention as the optimal approach, primarily due to the technical simplicity of surgery in this region, where primary anastomosis is often feasible. Additionally, the lower clinical success rates associated with stent placement in the right colon further support surgical management [16].

However, as evidence of successful outcomes with right-sided stenting accumulated, the indications for its use in proximal obstructions have expanded. Recent studies demonstrate that SEMS placement in the right colon yields outcomes comparable to those observed in the left colon without an increased risk of AEs, such as perforation [17,30,31,44] (Table 2). Consequently, right-sided stent placement is now considered a safe and effective approach, in both the BTS and palliative settings, being endorsed by the 2020 ESGE, 2021 AGA and the 2022 American Society of Colon and Rectal Surgeons guidelines [8,11,16].

## 6. Extracolonic Obstruction

Regarding SEMS placement in cases of extracolonic obstruction, it is considered a valid option, despite being associated with lower technical and clinical success rates compared to stenting in primary colonic cancer.

It is important to note that the procedure is often technically more challenging, with a higher incidence of AEs including stent migration and perforation. However, stenting remains a critical tool in this setting, especially when surgery is not feasible. Despite the higher risk of AEs, stenting can still provide significant palliative benefits, and therefore the decision should be carefully considered, weighing the potential risks against the expected benefits in individual cases [7,11,45].

## 7. Technical Considerations

SEMS placement in colonic obstructions should be performed by an experienced interventional endoscopist, as the success of the procedure is highly dependent on personal expertise [36,46]. Establishing a precise minimum number of stent placements to ensure proficiency remains challenging, as optimal numbers vary across sites based upon endoscopic capacity and patient volume; however, a minimum of 20 colonic stents, or experience in ERCP stenting and fluoroscopy understanding, should be required before starting placing stents independently [7,47].

Before the procedure, cleansing enemas should be used to clean the colon distal to the stricture, as it can make the procedure easier and faster, providing a clearer view of the obstruction to facilitate stent placement [7,48]. Routine antibiotic prophylaxis is generally not advised, since the risk of fever and bacteremia post-stent insertion is low. Instead, careful attention to procedural technique and post-procedure monitoring is essential to minimize any potential complications [49].

Colonic stenting can be performed using either the through-the-scope (TTS) or the over-the-wire (OTW) technique. Although conflicting results exist between the two techniques [7], it is common to adopt the TTS technique, as it allows a more controlled deployment, even though larger operative channels (>3.7 mm) are required. These procedures should always be undertaken in rooms with fluoroscopy support [7], even though in situations where this is not available, ultrathin scopes may be used to pass through the stricture, allowing safe guidewire placement, while adding additional information on upstream mucosal viability. While it is ideal to confirm malignancy through endoscopic biopsies during colonic stenting, this should only be attempted if it does not compromise the technical success of the procedure [7]. The use of double-channel endoscopes may facilitate it, with biopsies being performed through one channel of the endoscope while access to the stricture with a guidewire is ensured on the other channel.

After the correct characterization of the stricture with contrast injection, a long guidewire usually inserted through a cannula (or sphincterotome in the case of angled strictures) is passed through the stricture and allowed to loop several times proximal to the obstruction, and afterwards the cannula is advanced (ERCP technique). Stricture dilation should never be performed due to its significantly higher risk of perforation and no improvement in technical or clinical success [7]. Before advancing the stent, the correct location of the guidewire in the colon proximal to the obstruction should always be confirmed. This can be done by the additional injection of contrast or changing the patient’s position. If in doubt, remove the guidewire and restart the procedure. The stent delivery system is then advanced over the guidewire under radiologic and endoscopic visualization. Gentle traction should be applied to the guidewire without pulling it back, ensuring stability and control. Throughout stent deployment, the delivery system should be constantly pulled back to counterbalance the stent’s position and ensure accurate placement, as the tendency is to push the stent forward during its deployment [50,51].

Regarding stent selection, uncovered SEMSs are usually preferred. Studies show that both types (covered and uncovered) have similar technical and clinical success rates, with no significant differences in complications such as perforation or bleeding. However, uncovered SEMS are associated with lower rates of stent migration and tumor overgrowth, along with longer stent patency [52]. Usually, 22 to 24 mm stent body diameters are used, as these allow adequate expansion, with good patency rates. The length of the stent should be carefully tailored to match the length of the stricture. It is recommended that the stent extend beyond the stricture by 1.5 to 2 cm on both ends, while accounting for the degree of foreshortening after stent deployment. When in doubt, favor longer stents, as they may allow for better conformability to the stricture, especially when located in flexures. While there are various stent designs available, studies have shown no significant differences in efficacy and safety between different designs [7,12,35].

SEMS may be more challenging to deploy in specific anatomic locations, like the hepatic and splenic flexures, or the sigmoid/descending transition. In these situations, longer stents should be considered, to allow for the correct alignment of the stent with the bowel axis and prevent early stent dysfunction or AE occurrence.

## 8. Adverse Events

SEMS placement is associated with several potential AEs, with different degrees of severity. While the overall AE rate is reported to be 20 to 30%, the most concerning AE is perforation [7], which may occur in up to 5% of cases. Factors contributing to immediate perforation include guidewire/catheter misplacement or stricture dilation, usually at the level of the stricture, while delayed perforation may occur from excessive air insufflation, usually in the right colon, or from excessive radial pressure from the stent in a weakened bowel wall. Surgery is almost always the therapeutic of choice if perforation occurs.

Other AEs include stent migration (4–10%), which usually occurs during stent deployment, and stent obstruction, which may occur from tumor ingrowth, overgrowth, or stool impaction. Re-obstruction rates can be up to 30%. In addition to these major complications, other adverse effects such as ulceration, minor bleeding, infection and tenesmus can also occur [12,14,35]. In the palliative setting, if stent obstruction occurs, endoscopic re-intervention with stent-in-stent placement is recommended, while stent replacement is advised in cases of stent migration. In the curative setting, surgery is indicated if stent obstruction or migration occurs [7,37]. Bleeding, which may occur after stent placement, is usually minor and self-limited.

## 9. Discussion

The use of SEMSs in the management of CRC, particularly in cases of MCO, has evolved significantly over the past decade. This review provides a comprehensive analysis of the latest evidence regarding indications, contraindications and outcomes associated with stent placement, both as a BTS and in palliative settings.

In the curative context, BST stenting offers short-term advantages such as reduced morbidity and lower stoma formation rates. Despite previous concerns about oncological outcomes, recent studies show long-term outcomes comparable to those of ES. In palliative care, colonic stenting is widely recommended as it effectively relieves symptoms and enhances the quality of life, despite the potential for long-term AEs in patients with extended survival. Stenting in the proximal colon, previously less favored due to technical challenges, is now considered a safe and effective option with outcomes comparable to left-sided stenting, both as a BTS and for palliative care. SEMS placement for extracolonic obstruction is also a viable option, although it carries higher risks and technical challenges.

Technical expertise, careful procedural planning, and appropriate stent selection are essential to reduce the risk of AEs, such as perforation, migration and stent obstruction.

This review has some limitations that should be considered. A primary issue is the heterogeneity among the included studies, with significant variations in methodologies, patient demographics, and outcome measures, making consistent comparisons challenging. Additionally, the long-term oncological outcomes of SEMS use remain uncertain, with ongoing debates regarding its influence on recurrence and survival. We also highlight the still-insufficient data regarding the safety of combining SEMS placement with neoadjuvant and adjuvant chemotherapy, particularly concerning risks of stent-related AEs. Furthermore, the procedural success and patient outcomes are highly dependent on the operator’s expertise, limiting the broader applicability of the findings. Lastly, the reliance on retrospective studies and the limited availability of large-scale randomized controlled trials underscore the necessity for further rigorous research to validate these conclusions.

Future research should optimize SEMS use in MCO by addressing patient selection, surgery timing, and long-term outcomes. Key areas include assessing neoadjuvant chemotherapy’s impact, expanding SEMS use in proximal obstructions, and improving stent design to reduce AEs. Studies on SEMSs’ interactions with angiogenic agents are also needed. Enhanced multidisciplinary approaches and training will support better implementation and outcomes, guiding future clinical guidelines.

## 10. Conclusions

Overall, colonic stenting is an important tool in managing obstructive CRC, providing significant immediate and long-term benefits when used appropriately. Ongoing research and clinical trials will further refine the SEMS application, helping to ensure the best possible outcomes for patients.

## Figures and Tables

**Figure 1 cancers-17-00087-f001:**
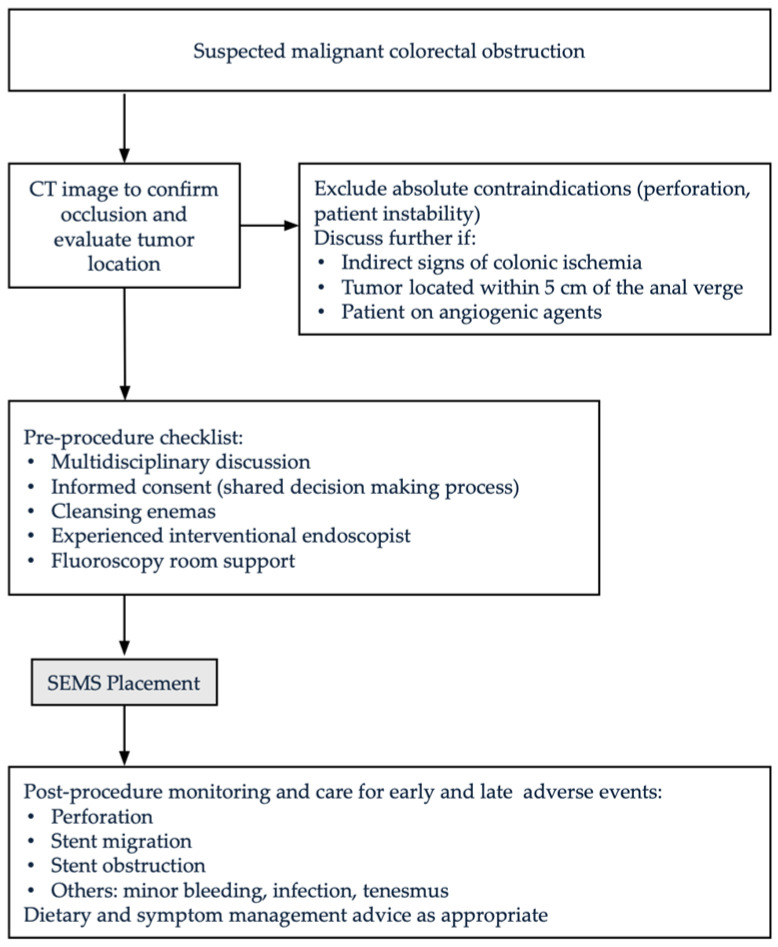
Visual algorithm illustrating key decision-making points and essential considerations at different stages of the process.

**Table 1 cancers-17-00087-t001:** Recent studies regarding outcomes of self-expandable metal stent (SEMS) placement in left malignant colorectal obstruction.

First Author, Year	Aim	Study Population	Results
Arezzo, A. 2020 [20]	Evaluate OS, time to progression and disease-free survival at 3 years after treatment for left-sided MCO (SEMS vs. ES)	115 patients (BTS 56 vs. ES 59)	- No differences were observed in the BTS group vs. ES in terms of OS (HR 0.93, _95%_CI 0.49–1.76, *p* = 0.822), time to progression (HR 0.81, _95%_CI 0.42–1.54, *p* = 0.512), and disease-free survival (HR 1.01, _95%_CI 0.56–1.81, *p* = 0.972)
Hill, J. 2022 [21]	Compare outcomes of stenting followed by surgery vs. surgical decompression with or without tumor resection for left-sided MCO	245 patients (123 SEMS vs. 122 ES)	- Technical and clinical success rates of 96.7% and 82.4%, respectively - Curative intent: no significant differences in 30-day postoperative mortality (3.6% vs. 5.6%, *p* = 0.48) or duration of hospital stay (median 19 [IQR 11–34] vs. 18 [10,11,12,13,14,15,16,17,18,19,20,21,22,23,24,25,26,27,28] days; *p* = 0.94) - Potentially curative treatment: stoma formation occurred less frequently in SEMS vs. ES (47.5% vs. 67.9%, *p* = 0.003); no significant differences in perioperative morbidity, critical care use, quality of life, 3-year recurrence or mortality between treatment groups
Ouyang, K. 2023 [23]	Evaluate the best treatment for left-sided MCO (ES vs. SEMS vs. TD vs. DS)	Meta-analysis (51 articles)	- More permanent stoma in ES and TD groups than in SEMS and DS groups [OR (_95%_CI) TD vs. SEMS: 4.12 (1.89–9.45); TD vs. DS: 3.39 (1.46–8.75); ES vs. DS: 2.55 (1.73–4.17); SEMS vs. ES: 0.33 (0.24–0.42)] - More morbidity in ES group than in SEMS group (ES vs. SEMS: OR 1.44, _95%_CI 1.14–1.82) - SEMS was ranked first in avoiding infection (Pro-4 0.95) - For in-hospital mortality, ES was ranked first (Pro-1 0.93) - TD was ranked first in recurrence (Pro-1 0.97) and metastasis (Pro-1 0.98) - No differences in 5-year OS and disease-free survival among all strategies
Han, J.G. 2023 [28]	Role of neoadjuvant chemotherapy during stent therapy in the curative setting in left-sided MCO	100 patients (52 stent then surgery vs. 48 stent, neoadjuvant chemotherapy then surgery)	- No statistically significant differences regarding stent-related AEs - Laparoscopic surgery was performed more frequently (77.1% vs. 40.4%; *p* < 0.001) and a stoma was created less frequently (10.4% vs. 28.8%; *p* = 0.02) in the chemotherapy group

AEs: adverse events. BTS: bridge-to-surgery. CI: confidence interval. DS: decompressive stoma. ES: emergency surgery. HR: hazard ratio. MCO: malignant colorectal obstruction. OS: overall survival. SEMS: self-expandable metal stents. TD: transanal drainage tube.

**Table 2 cancers-17-00087-t002:** Recent studies regarding outcomes of self-expandable metal stent (SEMS) placement in left- and right-sided malignant colorectal obstruction and chemotherapy use in this context.

First Author, Year	Aim	Study Population	Results
Liam Spannenburg, 2020 [18]	Compare surgical and oncological outcomes between ES and BTS for MCO	27 studies with a total of 3894 patients	- No significant differences in 3- and 5-year disease-free and overall survival - Stenting resulted in less blood loss (mean difference −234.72, *p* < 0.00001) and higher primary anastomosis rate (RR 1.25, *p* < 0.00001) - In the BTS setting, stenting was associated with a lower 30-day mortality rate (RR 0.65, *p* = 0.01), lower overall complication rate (RR 0.65, *p* < 0.0001), more lymph nodes harvested (mean difference 2.51, *p* = 0.005), shorter ICU stay (mean difference −2.27, *p* = 0.02) and shorter hospital stay (mean difference −7.24, *p* < 0.0001)
Kuwai, T. 2022 [22]	Evaluate long-term outcomes of BTS using standardized SEMS placement	208 patients	Short - term outcomes - Technical and clinical success rates of 99.0% and 92.8%, respectively - Laparoscopic and open surgeries in 62.0% and 33.7%, respectively - Colectomy with primary anastomosis in 92.8% Long - term outcomes - The 1-, 3-, and 5-year OS rates were 94.1%, 77.4%, and 67.4%, respectively (mean follow-up period of 38.8 ± 18.6 months) - OS rates of patients with stage II and III tumors were 95.8% and 92.8% at 1 year, 88.2% and 68.3% at 3 years, and 81.2% and 55.6% at 5 years, respectively (*p* = 0.0008).
Matsuda, A. 2024 [29]	Role of adjuvant chemotherapy (Adj) during stent therapy in the curative setting (BTS)	Adj vs. No-Adj: 77 vs. 52 patients	- RFS rates at 3 years were significantly different between the No-Adj and Adj groups (56.4% vs. 78.5%, *p* = 0.003) - Significant RFS benefits of Adj were observed in both pathological stage II and III cancer - Evidence of a survival benefit of Adj in patients with MCO undergoing BTS using a SEMS
Huang, Y. 2024 [17]	Compare outcomes of BTS vs. ES for right-sided MCO (SEMS placement without fluoroscopic assistance)	95 patients (35 BTS vs. 60 ES)	- Technical and clinical success rates were 100% and 88.6%, respectively - AEs: 1 case of stent-related perforation, 1 case of stent displacement
Oh, H.H. 2022 [30]	Compare outcomes of palliative SEMS placement in right-sided MCO vs. left-sided MCO	469 patients (69 right-sided vs. 400 left-sided)	Right-sided MCO vs. Left-sided MCO - Clinical success rates: 97.1% vs. 88.2% - AEs including stent migration, tumor ingrowth, outgrowth, perforation, bacteremia/fever, and bleeding: 10.1% vs. 19.9% - Mean overall survival: 28.02 months vs. 18.23 months - Palliative SEMS placement in right-sided MCO showed better clinical success rates than left-sided MCO
Kanaka, S.; 2022 [31]	Compare short-term outcomes between BTS and ES for right-side MCO	7 studies including 5136 patients (33% BTS and 67% ES)	- BTS with fewer postoperative complications (OR = 0.78, _95%_CI: 0.66–0.92) and mortality (OR = 0.51, _95%_CI: 0.28–0.92) than ES - Postoperative mortality rates in the BTS and ES groups were 0.9% and 5.2%, respectively - Rates of primary anastomosis favored the BTS group over ES group (97.8% and 85.9%, respectively), and stoma construction was preferable in the BTS group (2.0% and 11.0%, respectively) - Rates of anastomotic leakage and surgical site infection demonstrated significantly favorable results in the BTS group over the ER group (anastomotic leakage: OR = 0.66, _95%_CI 0.45–0.96; surgical site infection: OR = 0.62, 95% CI 0.46–0.82)

AEs: adverse events. BTS: bridge-to-surgery. CI: confidence interval. ES: emergency surgery. ICU: intensive care unit. MCO: malignant colorectal obstruction. OS: overall survival. RFS: relapse-free survival. RR: relative risk. SEMS: self-expandable metal stents.

## Data Availability

No new data were created or analyzed in this study. Data sharing is not applicable to this article.

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
