# Peer review of "Self-Expandable Metal Stents for Obstructing Colon Cancer and Extracolonic Cancer: A Review of Latest Evidence"

_cancers, 2024, doi:10.3390/cancers17010087_

Round 1
Reviewer 1 Report
Comments and Suggestions for Authors
This is a well written review of the use of self expanding metal stents for malignant colorectal cancer obstruction. I have 2 small corrections:
Line 149: 3.1. Safety rof Chemotherapy During Stent Therapy in the Curative Setting. . "rof" should be "of"
Lines 166-167: < 0.001) and a reduced need 166 for stoma “confection.” Do you mean stoma creation?
Author Response
Comment 1: “This is a well written review of the use of self expanding metal stents for malignant colorectal cancer obstruction. I have 2 small corrections:
Line 149: 3.1. Safety rof Chemotherapy During Stent Therapy in the Curative Setting. . "rof" should be "of"
Response 1: Thank you for pointing this out. Indeed, it should be “of”. We made this correction in the manuscript (page 6).
Comment 2: “Lines 166-167: < 0.001) and a reduced need for stoma “confection.” Do you mean stoma creation?
Response 2: Thank you for pointing this out. Yes, we meant stoma creation. We made this correction in the manuscript (page 6).
Reviewer 2 Report
Comments and Suggestions for Authors
The authors have done a good job at presenting a comprehensive literature review on a topic that is relevant to the readers of our journal. The following queries must be addressed:
1. The language is not consistent in terms of British vs American English. For example, "randomised" and "tumor." Please select one type of English according to our journal's specifications and preference and remain consistent.
2. The authors must do a better job at emphasizing the revelance of their contribution to the literature in the introduction. If this is a novel type of literature review, they must state it so. If not novel, how does this review contribute to the existing literature? What is its relevance?
3. The table (Table 1) is outstanding, yet very long and populated with a lot of information that must be summarized. Otherwise, it reads as if were a group of paragraphs together. While it is an excellent table, it must be abbreviated to avoid confusion and distraction in the readers.
4. Within the section on technical considerations, the authors should include a few sentences or at least a paragraph on the technique to deploy the stents and challenging scenarios. It is a liteature review, yes, but it does not mean that a technical description should be omitted. The readers are looking for that, too.
5. A paragraph on limitations of the study is lacking in the discussion.
6. A few sentences on future directions and ideas for further reserach are needed at the end of the discussion.
7. Can the authors please provide a diagram with an algorithm, original work, for the readers to understand the indications and caveats associated with the use and deployment of stents in a graphic fashion, for visual learners? It would enrich their manuscript.
8. The conclusion is very long and contains a lot of information that belongs in the discussion. It should be summarized, please.
Congratulations to the authors. I look forward to reviewing the revised manuscript if the editor in chief agrees to proceed with that phase.
Author Response
Comment 1: “The language is not consistent in terms of British vs American English. For example, "randomised" and "tumor." Please select one type of English according to our journal's specifications and preference and remain consistent.”
Response 1: Thank you for pointing this out. We agree with this comment. Therefore, we have changed all manuscript to American English. Please find the updated text in the resubmitted manuscript (all changes are highlighted).
Comment 2: “The authors must do a better job at emphasizing the relevance of their contribution to the literature in the introduction. If this is a novel type of literature review, they must state it so. If not novel, how does this review contribute to the existing literature? What is its relevance?”
Response 2: Thank you for your insightful comment. This review provides a comprehensive and updated overview of the use of SEMS in the management of malignant colorectal obstruction. While several descriptive studies and reviews exist, many are either focused on specific aspects of SEMS placement or are embedded within broader guideline documents. The most recent guidelines by major societies, such as ESGE (2020), provide valuable recommendations but do not integrate findings from the most recent trials and systematic reviews published since then. Our objective was to consolidate the latest evidence from recent high-quality studies and meta-analyses, reflecting advancements in SEMS application, particularly concerning its oncological safety, use in challenging clinical contexts (e.g., right-sided obstructions, extracolonic involvement), and technical considerations. Although this is not a novel type of literature review, it fills an important gap by offering a timely synthesis of the latest knowledge in this rapidly evolving area, which has not been structured in a comprehensive manner to date. We hope this clarification emphasizes the relevance of our work to the current body of literature. We reformulated the introduction accordingly (page 2).
Comment 3: “The table (Table 1) is outstanding, yet very long and populated with a lot of information that must be summarized. Otherwise, it reads as if were a group of paragraphs together. While it is an excellent table, it must be abbreviated to avoid confusion and distraction in the readers.”
Response 3: Thank you for your feedback on Table 1. We created this table to summarize the key studies in SEMS placement in left and right-sided malignant colorectal obstruction, chemotherapy use in this context, and to present their main conclusions in a clear and organized manner. We understand your concern regarding its length and potential impact on readability. To address this, we reformulated and summarized it and divided it in two tables, one focusing in studies concerning left sided MCO and the other for studies that included both sides MCO and other situations. We also simplified the table by removing the column related to the title. Please find these changes in the new reformulated tables 1 and 2 (page 4 and 5).
Comment 4: “Within the section on technical considerations, the authors should include a few sentences or at least a paragraph on the technique to deploy the stents and challenging scenarios. It is a literature review, yes, but it does not mean that a technical description should be omitted. The readers are looking for that, too.
Response 4: Thank you for your insightful comment. We have added information regarding challenging scenarios as suggested. We hope these additions adequately address your concerns and enhance the manuscript.
Comment 5: “A paragraph on limitations of the study is lacking in the discussion.”
Response 5: We appreciate your observation. To address this, we have added a dedicated paragraph in the discussion section outlining the study's limitations.
Comment 6: “A few sentences on future directions and ideas for further research are needed at the end of the discussion.”
Response: 6: Thank you for your suggestion. Therefore, we have added a paragraph about future directions in the discussion section. This includes recommendations for exploring optimal surgery timing after SEMS placement, evaluating the role of neoadjuvant chemotherapy, expanding SEMS use in challenging cases like proximal and extracolonic obstruction, and advancing stent designs to reduce complications. These additions are made in the "Discussion" section on page 10. Please let us know if further modifications are needed.
Comment 7: “Can the authors please provide a diagram with an algorithm, original work, for the readers to understand the indications and caveats associated with the use and deployment of stents in a graphic fashion, for visual learners? It would enrich their manuscript.”
Response 7: Thank you for your suggestion. We agree that a diagram illustrating the indications and caveats for stent use would enhance the manuscript. Accordingly, we have created an original algorithm to provide a step-by-step guide for the management of MCO using SEMS. We hope this addition enriches the manuscript and meets the reviewer's expectations.
Comment 1: “The conclusion is very long and contains a lot of information that belongs in the discussion. It should be summarized, please.”
Response 1: Thank you for your observation. To address this, we have restructured the manuscript by integrating the details from the conclusion into the discussion section. We have expanded the discussion to include limitations of our review and suggestions for future research directions. We shortened the conclusion to provide a concise overview of the key findings and take-home messages of the study. These changes can be found in page 9 and 10.
Reviewer 3 Report
Comments and Suggestions for Authors
Please make corrections as attached

Need little English language corrections
Author Response
Thank you for the various corrections throughout the text. They were indeed helpful, and we corrected most the changes suggested. It is worth mentioning that regarding the consistency of writing between US English and UK English, we have chosen to use US English.
Reviewer 4 Report
Comments and Suggestions for Authors
This is a comprehensive review which systematically discusses the use of self-expandable stents in managing colorectal obstruction due to cancer disease. The article is very well balanced and provides excellent overview on this subject. I have only one major criticism: the title of the section 4.1 “Safety of Adjuvant Therapy in the Palliative Setting” is actually wrong: the term “adjuvant therapy” describes antitumor therapy given after the surgery, i.e. adjuvant therapy is aimed to eradicate invisible tumor cells after potentially curative intervention. Other remarks are minor and can be disregarded. The title of the paper mentions colon cancer, however, the authors also discuss management of extracolonic obstruction in the section 6. This is an advantage and the section 6 may be expanded. Perhaps, some words may be dedicated to the manufacturers of relevant devices. The text has to be checked with regard to abbreviations, as some of them are not explained.
Author Response
Comment: This is a comprehensive review which systematically discusses the use of selfexpandable stents in managing colorectal obstruction due to cancer disease. The article is very well balanced and provides excellent overview on this subject. I have only one major criticism: the title of the section 4.1 “Safety of Adjuvant Therapy in the Palliative Setting” is actually wrong: the term “adjuvant therapy” describes antitumor therapy given after the surgery, i.e. adjuvant therapy is aimed to eradicate invisible tumor cells after potentially curative intervention. Other remarks are minor and can be disregarded. The title of the paper mentions colon cancer, however, the authors also discuss management of extracolonic obstruction in the section 6. This is an advantage and the section 6 may be expanded. Perhaps, some words may be dedicated to the manufacturers of relevant devices. The text has to be checked with regard to abbreviations, as some of them are not explained.
Response: Thank you for your feedback and assessment of our work. We agree with your comment regarding the title of Section 4.1 and have corrected it to “Safety of Chemotherapy in the Palliative Setting” to ensure accuracy. Additionally, we carefully reviewed the manuscript to ensure all abbreviations are properly explained. Finally, we have revised the article title to better reflect its content.
Round 2
Reviewer 2 Report
Comments and Suggestions for Authors
The authors have answered the queries appropriately and improved their manuscript. I am pleased with the manuscript in its current form and approve.